# Spatially Resolved Meteorological and Ancillary Data in Central Europe for Rainfall Streamflow Modeling

Marc Aurel Vischer<sup>1</sup>, Noelia Otero<sup>1</sup>, and Jackie Ma<sup>1</sup>

<sup>1</sup>Fraunhofer Heinrich-Hertz Institute, Applied Machine Learning Group, 10587 Berlin, Germany

Correspondence: Marc Aurel Vischer (marc.aurel.vischer@hhi.fraunhofer.de) and Jackie Ma (jackie.ma@hhi.fraunhofer.de)

**Abstract.** We present a gridded dataset for rainfall streamflow modeling that is fully spatially resolved and covers five complete river basins in central Europe: upper Danube, Elbe, Oder, Rhine, and Weser. We compiled meteorological forcings and a variety of ancillary information on soil, rock, land cover, and orography. The data is harmonized to a regular  $9km \times 9km$  grid, temporal resolution is daily from 1980 to 2024. We also provide code to further combine our dataset with publicly available river discharge data for end-to-end rainfall streamflow modeling. We have used this data to demonstrate how neural network-driven hydrological modeling can be taken beyond lumped catchments, and want to facilitate direct comparisons between different model types.

## 1 Introduction

In recent years, a substantial number of rainfall streamflow datasets were released that follow the example of the popular CAMELS dataset (Newman et al., 2015; Addor et al., 2017). They cover Chile (Alvarez-Garreton et al., 2018), Great Britain (Coxon et al., 2020), Brazil (Chagas et al., 2020), Australia (Fowler et al., 2021), the upper Danube basin (Klingler et al., 2021), France (Delaigue et al., 2022), Switzerland (Höge et al., 2023), Denmark (Liu et al., 2024) and Germany (Loritz et al., 2024). These datasets bundle a range of data sources and harmonize them to a readily ingestible, common spatio-temporal data format. Besides meteorological variables, they contain additional static information such as land cover, soil and bedrock type and orographic features. While suitable for a range of hydrological modeling approaches, these publications specifically facilitated a surge in popularity of neural network models for rainfall streamflow modeling. For example, Kratzert et al. (2019); Nearing et al. (2024) have shown that neural network models are particularly suited to learn from such multi-variate, large-scale data.

A common downside of the above-mentioned datasets however is that they aggregate ("lump") each variable within a catchment to a single value. By doing so, all information about spatial variability is lost: A pattern of soil types might be reduced to the most prevalent one, or a range of different amounts of precipitation over a large area might be averaged to a single, unexpressive average value. This reduction of information is unnecessary and counter-intuitive, especially for large catchments or catchments with high spatial variability. The principle advantage of spatially resolved inputs is that they enable the model to capture spatial covariance among different variables, e.g. the interacting effects of soil sealing or steepness of terrain and a torrential rainfall. Physical models, still the standard model type in active operation, resolve their equations on such a grid

https://doi.org/10.5194/essd-2025-556

Preprint. Discussion started: 11 November 2025

© Author(s) 2025. CC BY 4.0 License.

for exactly this reason, but neural network training also benefits from vast amounts of data. Additionally, as each point on the grid contains a complete, self-contained set of meteorological and ancillary variables, the grid locations can be processed independently.

## 2 Methods

The study area of the dataset covers 5 entire basins in central Europe, namely the upper reaches of the Danube (until Bratislava), Elbe, Oder, Rhine and Weser. It is contiguous, 570.592 km<sup>2</sup> large and spans 10 countries. The temporal coverage ranges from 1 January 1980 to 31 December 2024. We bundle 6 spatiotemporal ("dynamic") meteorological features with 46 static ("ancillary") features: 3 hydro-geological features, 16 land cover features, 19 soil features and 8 orographical features. We based our choice of which kind of dynamic and ancillary information to include on the work of Addor et al. (2017) and Kratzert et al. (2019) to allow for maximum comparability with recent hydrological literature in general and neural networkbased literature in particular. The dataset consists of data derived from a variety of publicly available sources - no new data was recorded. Our contribution consists in collecting the data and harmonizing it to a common grid for convenient model training. Figure 1 provides an overview of the study area, common grid and types of variables. The remainder of this section explains how the different data sources were harmonized to a common grid, followed by a description of each data modality. Original data sources are listed in Table A1, detailed lists of all dynamic and ancillary features that we derived to compile this dataset can be found in Tables A2 and A3. Along with the data, we release all scripts for processing the raw source data into the dataset that we provide. This allows users to verify and adapt our data aggregation pipeline. We also provide an additional script that combines the dataset presented here with river discharge data, after manual download from the the original provider, the Global Runoff Data Center (GRDC). In the preprocessing code linked below, we show that our study area is covered densely and uniformly with river gauging stations, making it suitable for training neural networks and other models. As there are much fewer stations in the lower Danube basin, we decided to only include the upper part in order to reduce sampling bias. The discharge time series come at daily resolution, which is the reason that we provide our temporal features in daily resolution as well. This data can serve as targets for end-to-end training in data-driven rainfall streamflow modeling, such as in our study (Vischer et al., 2025b).

### 2.1 Harmonized Grid

As a common grid format, we decided to use the grid of the ERA5-Land reanalysis dataset (Muñoz Sabater, 2019; CopernicusClimateChangeService, 2022), which covers the earth's surface with a  $0.1^{\circ} \times 0.1^{\circ}$  resolution, corresponding to roughly  $9km \times 9km$  for the case of our study area. ERA5-Land contains a vast number of meteorological variables and has an hourly resolution, spanning from 1950 onward. It has been widely used and is actively maintained and updated. This means the dataset we provide with this paper remains easily extendable, should a user like to e.g. include an additional meteorological variable in their experiments, extend the study area or increase the temporal resolution. The spatial data originally comes in different formats (vector or grid), projections and resolutions. All data sources were thus harmonized to the grid of ERA5-Land by

**Figure 1.** Overview of study area, input grid and data types. (a) Study Area: The study area comprises 5 basins that cover a contiguous area in central Europe. (b) Input Grid and Station Network in Upper Danube Basin: Cells of input grid (orange) for Upper Danube basin. Catchment boundaries (black) are overlaid with corresponding stations (blue), as well as connecting arrows representing the station connectivity network. Code to reproduce the river network is released together with this paper. (c) Input Types: Visualizations for one example feature of each type of input. Basin outlines (black) and borders of Germany (turquoise) are plotted for reference.

Orography in panel (a) was adapted from the European Space Agency's Copernicus Global 90 m DEM (GLO-90, doi:10.5270/ESA-c5d3d65) © EuroGeographics for the administrative boundaries in panel (a) and (c). Watershed boundaries in panel (a), (b), and (c) were taken from the Global Runoff Data Center.

means of reprojecting and subsampling at the locations of the nodes in this grid. Our study area comprises a total of 7169 grid cells.

# 60 2.2 Meteorological Forcings

The meteorological forcings in our study were derived from the ERA5-Land dataset<sup>1</sup> (Muñoz Sabater, 2019; CopernicusClimateChangeService, 2022). Temporal aggregation from hourly to daily resolution was achieved differently for each variable:

<sup>&</sup>lt;sup>1</sup>The dataset was downloaded from the Copernicus Climate Change Service (2022). The results contain modified Copernicus Climate Change Service information 2020. Neither the European Commission nor ECMWF is responsible for any use that may be made of the Copernicus information or data it contains.

https://doi.org/10.5194/essd-2025-556

Preprint. Discussion started: 11 November 2025

© Author(s) 2025. CC BY 4.0 License.

The temperature two meters above surface was aggregated by calculating minimum, mean and maximum values. Potential evapotranspiration was summed. Precipitation is provided in ERA5-Land as sub-daily values, meaning that the daily total sum corresponds to the value stored at 24:00. We added a measure of variability of precipitation by taking the variance over the increment at every hourly time step. Table A2 provides a detailed list of all dynamic variables.

# 2.3 Ancillary Data

Hydrogeological properties were derived from the International Hydrogeological Map of Europe (IHME) <sup>2</sup> (Günther and Duscher, 2019). The original dataset features six hydrogeological classes as well as two classes for snow-ice-fields and inland water bodies. The six classes represent the productivity of rock type, which indicates how easily water can dissipate through the bedrock. Classes are ordinal in that they are sorted by the corresponding productivity in ascending order. This allows us to take a non-rigorously defined but nonetheless informative average over the classes' proportions within each grid cell. We concatenate this productivity score with the binary categorical classes for snow-ice-fields and inland water bodies, each represented by a ratio of prevalence of this type of binary class within the grid cell.

Land Cover information was obtained from the Corine Land Cover Map<sup>3</sup> (CLC). This dataset classifies land cover at three different levels of detail, with increasingly differentiated (sub)classes. We decided to use the second level, which containing 16 classes in total. Similarly to the procedure applied to the hydrogeological properties, we calculated a distributional vector representing the proportion of a given class covering the grid cell.

Soil type information was obtained from the dataset European Soil Database Derived Data <sup>4</sup> (Hiederer, 2013a, b). This dataset features 17 different physical properties, separately for top soil and lower soil. We calculate the average value of each feature within a grid cell.

*Orographic information* was derived from the European Union Digital Elevation Map<sup>5</sup> (EU-DEM). Elevation was averaged within each grid cell, as well as the gradient in latitudinal and longitudinal direction, and the steepness or magnitude of the two-dimensional gradient. This yielded a total of four orographic features.

Table A3 provides an detailed list of all ancillary variables in the same ordering as we just introduced, which is also the ordering in the data file.

<sup>&</sup>lt;sup>2</sup>IHME1500 - Internationale Hydrogeologische Karte von Europa 1:1.500.000, version 1.2 © Bundesanstalt für Geowissenschaft und Rohstoffe, 2022.

<sup>&</sup>lt;sup>3</sup>Corine Land Cover Map, version 2012. Generated using European Union's Copernicus Land Monitoring Service information; https://doi.org/10.2909/916c0ee7-9711-4996-9876-95ea45ce1d27. The Corine Land Cover Map data was created with funding by the European union. It was adapted and modified by the authors.

<sup>&</sup>lt;sup>4</sup>European Soil Database Derived Data, created by the European Soil Data Centre with funding by the European union. It was adapted and modified by the authors. The authors' activities are not officially endorsed by the Union.

<sup>&</sup>lt;sup>5</sup>European Union Digital Elevation Map, version 1.1. Generated using European Union's Copernicus Land Monitoring Service information. The European Union Digital Elevation Map created with funding by the European union. It was adapted and modified by the authors. The authors' activities are not officially endorsed by the Union.

https://doi.org/10.5194/essd-2025-556

Preprint. Discussion started: 11 November 2025

© Author(s) 2025. CC BY 4.0 License.

100

110

#### 3 Conclusions

Combining a variety of data sources, we provide a dataset that is suitable for spatially resolved, multivariate rainfall-streamflow modeling at large scale and of entire river basins. Recent advances in computer memory have made parallel processing of such data without prior aggregation practically feasible: In Vischer et al. (2025b), we show that a neural network model is capable of efficiently handling this large amount of data. With the publication of this dataset, we hope to stimulate further development of hydrological, particularly network models beyond the scope of lumped catchments, as well as facilitate comparisons between different modeling approaches.

Code and data availability. The dataset is available at https://doi.org/10.4211/hs.d7f2cbb587ab4a75ac7987854e8f62ca, (Vischer et al., 2025a). Dynamic meteorological forcing data and static ancillary data are stored in two separate NetCDF4 (Rew et al., 2006) files, "ancillary\_pub.nc" and "dynamic\_pub.nc". This format allows for labeled coordinates such as latitude, longitude and date for convenient selection on spatial and temporal domains, respectively. All variables are named in a self-explanatory manner and we provide labeled metadata. See Tables A2 and A3 for a detailed reference of the included variables.

The data was processed in several Python Jupyter Notebooks (Granger and Pérez, 2021) that can be found in this repository. The code requires Python 3.11 (Van Rossum and Drake Jr, 2009) and is licensed under the Clear BSD licence. Additional dependencies are specified in an Anaconda environment (Anaconda, 2020) specification contained in the repository. The scripts are stand-alone and do not require further input parameters. Along with the code to process the data, we provide a script that loads the data, selects subsets and visualizes them. This can serve as a starting point for the user to interact with the data. Furthermore, we provide code to wrap all the data in a PyTorch (Ansel et al., 2024) Dataset class for further processing in a machine learning context. Since dense arrays are required for this, we provide an alternative format version of our features in the files "ancillary\_paper.nc" and "dynamic\_paper.nc", where dimensions were transformed from latitude and longitude to a unique grid cell index. In this version, all features were standardnormalized in order to suit better the requirements of neural networks.

All data sources from which we obtained the original data have been widely used across various scientific fields for years, so we assume the original data to be valid. In order to technically validate our processing steps, we feature a testing script in our repository with extensive tests and visualizations of the compiled data. We also successfully employed this dataset in training a neural network model for rainfall streamflow modeling (Vischer et al., 2025b).

# Appendix A: Data Origins and Detailed List of Features

**Table A1.** Overview of source datasets and their authors for dynamic data / meteorological forcings contained in file *forcings\_pub.nc* and static / ancillary data contained in *ancillary\_pub.nc*. See tables A2 and A3 for more details on derived features.

| Туре                           | Dataset                                                    | Author                                                               | Citation                              |
|--------------------------------|------------------------------------------------------------|----------------------------------------------------------------------|---------------------------------------|
| Forcings / Dynamic Inputs      |                                                            |                                                                      |                                       |
| Meteorological Variables       | ERA5-Land                                                  | Copernicus Climate Change Service (CCCS)                             | Muñoz Sabater (2019)                  |
|                                |                                                            |                                                                      | CopernicusClimateChangeService (2022) |
| Ancillary Data / Static Inputs |                                                            |                                                                      |                                       |
| Hydrogeological Properties     | IHME hydrogeological map v1.2 in vector data format        | German Federal Institute for Geosciences and Natural Resources (BGR) | Günther and Duscher (2019)            |
| Land Cover                     | Corine Land Cover Map, version 2012                        | Copernicus Land Monitoring Service (CLMS)                            |                                       |
| Soil Type (Top and Lower Soil) | European Soil Database Derived Data                        | European Soil Data Centre (ESDAC)                                    | Hiederer (2013a, b)                   |
| Orographic                     | European Union Digital Elevation Map (EU-DEM), version 1.1 | Copernicus Land Monitoring Service (CLMS)                            |                                       |

**Table A2.** Overview of dynamic input features in the file *forcings\_pub.nc*. Empty cells indicate that the value is identical to the one above. Each of these features is a three-dimensional array with dimensions latitude, longitude and date. Labeled coordinate indices for all dimensions are contained in the file.

| Index | Name     | Feature                    | Origin | Aggregation    | Unit     |
|-------|----------|----------------------------|--------|----------------|----------|
| 00    | t2m_min  | Temperature 2m above ERA5- |        | Daily Minimum  | K        |
|       |          | ground                     | Land   |                |          |
| 01    | t2m_mean |                            |        | Daily Mean     |          |
| 02    | t2m_max  |                            |        | Daily Maximum  |          |
| 03    | pev      | Potential evapotranspira-  |        | Daily Sum      | mm       |
|       |          | tion                       |        |                |          |
| 04    | tp_sum   | Precipitation              |        | Daily Sum      |          |
| 05    | tp_var   |                            |        | Daily Variance | unitless |
|       |          |                            |        |                |          |

**Table A3.** Overview of static input features in the file *ancillary\_pub.nc*. Empty cells indicate that the value is identical to the one above. Explanations of the features derived from Corine Land Cover map (CLC) and elevation map were omitted because the names are self-explanatory. Each of these features is a two-dimensional array with dimensions latitude and longitude. Labeled coordinate indices for all dimensions are contained in the file.

| Index | Name                                                                                | Feature                                   | Origin        | Aggregation            | Unit              |  |  |  |
|-------|-------------------------------------------------------------------------------------|-------------------------------------------|---------------|------------------------|-------------------|--|--|--|
| 00    | IHME_AQUIF_CODE                                                                     | Rock Productivity                         | IHME          | Averaged Classes       | untiless          |  |  |  |
| 01    | IHME_INLAND_WATER                                                                   | Inland Water Body                         |               | Fraction               | frac. area        |  |  |  |
| 02    | IHME_SNOW_ICE_FIELD Permanent Snow-Ice Field                                        |                                           |               |                        |                   |  |  |  |
| 03    | CLC_11_Artificial_surfaces_Urban_fabr                                               | ic                                        | CLC           |                        |                   |  |  |  |
| 04    | CLC_12_Artificial_surfaces_Industrial,_commercial_and_transport_units               |                                           |               |                        |                   |  |  |  |
| 05    | CLC_13_Artificial_surfaces_Mine,_dump_and_construction_sites                        |                                           |               |                        |                   |  |  |  |
| 06    | CLC_14_Artificial_surfaces_Artificial_non_agricultural_vegetated_areas              |                                           |               |                        |                   |  |  |  |
| 07    | CLC_21_Agricultural_areas_Arable_land                                               |                                           |               |                        |                   |  |  |  |
| 08    | CLC_22_Agricultural_areas_Permanent_crops                                           |                                           |               |                        |                   |  |  |  |
| 09    | CLC_23_Agricultural_areas_Pastures                                                  |                                           |               |                        |                   |  |  |  |
| 10    | CLC_24_Agricultural_areas_Heterogeneous_agricultural_areas                          |                                           |               |                        |                   |  |  |  |
| 11    | CLC_31_Forest_and_seminatural_areas_Forest                                          |                                           |               |                        |                   |  |  |  |
| 12    | CLC_32_Forest_and_seminatural_areas_Shrub_and_or_herbaceous_vegetation_associations |                                           |               |                        |                   |  |  |  |
| 13    | CLC_33_Forest_and_seminatural_areas_Open_spaces_with_little_or_no_vegetation_       |                                           |               |                        |                   |  |  |  |
| 14    | CLC_41_Wetlands_Inland_wetlands                                                     |                                           |               |                        |                   |  |  |  |
| 15    | CLC_42_Wetlands_Coastal_wetlands                                                    |                                           |               |                        |                   |  |  |  |
| 16    | CLC_51_Water_bodies_Inland_waters                                                   |                                           |               |                        |                   |  |  |  |
| 17    | CLC_51_Water_bodies_Marine_waters                                                   |                                           |               |                        |                   |  |  |  |
| 18    | CLC_No_data                                                                         |                                           |               |                        |                   |  |  |  |
| 19    | SOIL_STU_EU_S_SILT                                                                  | Subsoil: Silt Content                     | ESDAC         | Arithmetic Mean        | %                 |  |  |  |
| 20    | SOIL_STU_EU_T_SAND                                                                  | Topsoil: Sand Content                     | 1             |                        |                   |  |  |  |
| 21    | SOIL_SMU_EU_S_TAWC                                                                  | Subsoil: Total Availab                    | le Water Cont | ent (Pedotr. Rule)     | mm                |  |  |  |
| 22    | SOIL_SMU_EU_T_TAWC                                                                  | Topsoil: Total Availab                    | le Water Cont | ent (Pedotr. Rule)     |                   |  |  |  |
| 23    | SOIL_STU_EU_T_BD                                                                    | Topsoil: Bulk Density                     |               |                        | g/cm <sup>3</sup> |  |  |  |
| 24    | SOIL_STU_EU_T_TAWC                                                                  | Topsoil: Total Availab                    | le Water Cont | ent (Pedotr. Function) | mm                |  |  |  |
| 25    | SOIL_STU_EU_S_GRAVEL                                                                | Subsoil: Coarse Fragn                     | nents         |                        | %                 |  |  |  |
| 26    | SOIL_STU_EU_DEPTH_ROOTS                                                             | Depth Available to Ro                     | oots          |                        | cm                |  |  |  |
| 27    | SOIL_STU_EU_T_GRAVEL                                                                | Topsoil: Coarse Fragn                     | nents         |                        | %                 |  |  |  |
| 28    | SOIL_STU_EU_S_TEXT_CLS                                                              | Subsoil: Texture Class                    | 3             |                        | unitless          |  |  |  |
| 29    | SOIL_STU_EU_T_OC                                                                    | SOIL_STU_EU_T_OC Topsoil: Organic Content |               | %                      |                   |  |  |  |
| 30    | SOIL_STU_EU_S_SAND                                                                  | Subsoil: Sand Conten                      | t             |                        |                   |  |  |  |
| 31    | SOIL_STU_EU_T_CLAY                                                                  | Topsoil: Clay Content                     |               |                        |                   |  |  |  |
| 32    | SOIL_STU_EU_T_TEXT_CLS                                                              | Topsoil: Texture Class                    | 3             |                        | unitless          |  |  |  |
| 33    | SOIL_STU_EU_T_SILT                                                                  | Topsoil: Silt Content                     |               |                        | %                 |  |  |  |
| 34    | SOIL_STU_EU_S_BD                                                                    | Subsoil: Bulk Density                     |               |                        | g/cm <sup>3</sup> |  |  |  |
| 35    | SOIL_STU_EU_S_TAWC                                                                  | Subsoil: Total Availab                    | le Water Cont | ent (Pedotr. Function) | mm                |  |  |  |
| 36    | SOIL_STU_EU_S_OC                                                                    | Subsoil: Organic Carb                     | on Content    |                        | %                 |  |  |  |
| 37    | SOIL_STU_EU_S_CLAY                                                                  | Subsoil: Clay Content                     |               |                        | %                 |  |  |  |
| 38    | DEM_elevation_mean                                                                  |                                           | EU-DEM        |                        | m                 |  |  |  |
| 39    | DEM_grad_x_mean                                                                     |                                           |               |                        | 1/m               |  |  |  |
| 40    | DEM_grad_y_mean                                                                     |                                           |               |                        |                   |  |  |  |
| 41    | DEM_steepness_mean                                                                  |                                           |               |                        | 1/m <sup>2</sup>  |  |  |  |
| 42    | DEM_elevation_std                                                                   |                                           |               | Standard Deviation     | unitless          |  |  |  |
| 43    | DEM_grad_x_std                                                                      |                                           |               |                        |                   |  |  |  |
| 44    | DEM_grad_y_std                                                                      |                                           |               |                        |                   |  |  |  |
| 45    | DEM_steepness_std                                                                   |                                           |               |                        |                   |  |  |  |

https://doi.org/10.5194/essd-2025-556 Preprint. Discussion started: 11 November 2025

© Author(s) 2025. CC BY 4.0 License.

Author contributions. M.A.V. compiled the data with crucial suggestions from N.O., processed the data, and wrote the manuscript with significant contributions from J.M. All authors reviewed the manuscript.

115 Competing interests. The authors declare no competing interests.

Acknowledgements. This work was supported by the Federal Ministry for Economic Affairs and Climate Action (BMWK) as grant DAKI-FWS (01MK21009A), and by the European Union's Horizon Europe research and innovation program (EU Horizon Europe) project MedEWSa under grant agreement no. 101121192.

© Author(s) 2025. CC BY 4.0 License.

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

© Author(s) 2025. CC BY 4.0 License.

scape Attributes for 331 Catchments in Hydrologic Switzerland, Earth System Science Data, 15, 5755–5784, https://doi.org/10.5194/essd-15-5755-2023, 2023.

- Klingler, C., Schulz, K., and Herrnegger, M.: LamaH-CE: LArge-SaMple DAta for Hydrology and Environmental Sciences for Central Europe, Earth System Science Data, 13, 4529–4565, https://doi.org/10.5194/essd-13-4529-2021, 2021.
- Kratzert, F., Klotz, D., Herrnegger, M., Sampson, A. K., Hochreiter, S., and Nearing, G. S.: Toward Improved Predictions in Ungauged Basins: Exploiting the Power of Machine Learning, Water Resources Research, 55, 11 344–11 354, https://doi.org/10.1029/2019WR026065, 2019.
  - Liu, J., Koch, J., Stisen, S., Troldborg, L., Højberg, A. L., Thodsen, H., Hansen, M. F. T., and Schneider, R. J. M.: CAMELS-DK: Hydrometeorological Time Series and Landscape Attributes for 3330 Catchments in Denmark, Earth System Science Data Discussions, pp. 1–30, https://doi.org/10.5194/essd-2024-292, 2024.
- Loritz, R., Dolich, A., Acuña Espinoza, E., Ebeling, P., Guse, B., Götte, J., Hassler, S. K., Hauffe, C., Heidbüchel, I., Kiesel, J., Mälicke, M., Müller-Thomy, H., Stölzle, M., and Tarasova, L.: CAMELS-DE: Hydro-Meteorological Time Series and Attributes for 1555 Catchments in Germany, Earth System Science Data Discussions, pp. 1–30, https://doi.org/10.5194/essd-2024-318, 2024.
  - Muñoz Sabater, J.: ERA5-Land Hourly Data from 1950 to Present. Copernicus Climate Change Service (C3S) Climate Data Store (CDS). DOI: 10.24381/Cds.E2161bac (Accessed on 17-Sep-2024), 2019.
- Nearing, G., Cohen, D., Dube, V., Gauch, M., Gilon, O., Harrigan, S., Hassidim, A., Klotz, D., Kratzert, F., Metzger, A., Nevo, S., Pappenberger, F., Prudhomme, C., Shalev, G., Shenzis, S., Tekalign, T. Y., Weitzner, D., and Matias, Y.: Global Prediction of Extreme Floods in Ungauged Watersheds, Nature, 627, 559–563, https://doi.org/10.1038/s41586-024-07145-1, 2024.
  - Newman, A. J., Clark, M. P., Sampson, K., Wood, A., Hay, L. E., Bock, A., Viger, R. J., Blodgett, D., Brekke, L., Arnold, J. R., Hopson, T., and Duan, Q.: Development of a Large-Sample Watershed-Scale Hydrometeorological Data Set for the Contiguous USA: Data Set
- 175 Characteristics and Assessment of Regional Variability in Hydrologic Model Performance, Hydrology and Earth System Sciences, 19, 209–223, https://doi.org/10.5194/hess-19-209-2015, 2015.
  - Rew, R., Harnett, E., and Caron, J.: NetCDF-4: Software Implementing an Enhanced Data Model for the Geosciences, in: 22nd International Conference on Interactive Information Processing Systems for Meteorology, Oceanograph, and Hydrology, vol. 6, 2006.
  - Van Rossum, G. and Drake Jr, F. L.: Python 3 Reference Manual, Scotts Valley: CreateSpace, 2009.
- Vischer, M. A., Otero, N., and Ma, J.: Spatially Resolved Meteorological and Ancillary Data in Central Europe for Rainfall Streamflow Modeling, https://doi.org/10.4211/hs.d7f2cbb587ab4a75ac7987854e8f62ca, 2025a.
  - Vischer, M. A., Otero, N., and Ma, J.: Spatially Resolved Rainfall Streamflow Modeling in Central Europe, EGUsphere, https://doi.org/10.5194/egusphere-2024-3649, 2025b.