# Peer review of "Spatially Resolved Meteorological and Ancillary Data in Central Europe for Rainfall Streamflow Modeling"

_Earth System Science Data, 2025_

## Author Comment (AC1)

Dear Reviewer, we thank you very kindly for the constructive comments and time spent in reviewing the manuscript. We have carefully revised the manuscript according to the comments and suggestions. Below we provide a point-by-point response where we discuss the changes we have made to the manuscript in order make novelty and scientific contribution more clear, as well as provide more satisfying methodological detail."

- *R1. The manuscript aggregates variables from multiple publicly available data sources and harmonizes them onto the ERA5-Land grid. However, it does not clarify whether this integration process involves any methodological innovation, nor does it explain what added value the dataset provides compared with researchers independently processing the original data themselves. Without such clarification, the dataset's novelty appears limited.*

  A1. We do not use novel methods in harmonization, but instead rely on well established algorithms to guarantee correctness of the processed data. The added value of our contribution instead consists in adding a new variety of dataset (spatially resolved) to a family of datasets (bundled meteorological and surface data) that has seen increasing popularity over the last years. Our dataset allows researchers to focus on model development rather than having to invest a lot of time into data processing beforehand. At the same time, general-purpose and ready-to-use datasets alleviate to a certain degree the requirements in domain knowledge on the part of modelers. As the entry threshold is lowered, hopefully more diverse models and creative approaches will be brought in to tackle to problem as a community. Re-using datasets not only saves time but allows for immediate comparison between different modeling approaches. For a dataset to serve this purpose, apart from public license of data and code, the peer review process is an indispensable requirement. It ensures the dataset's correctness and overall quality of composition, building trust especially for researchers who are new to this domain. We also want to note here that our dataset is readily extendable, as we provide code for downloading additional variables from ERA5 incorporating them to the dataset. This way our dataset can serve as a starting point for different but adjacent modeling problems as well.

- *R2. The rationale for selecting the 6 spatiotemporal ("dynamic") meteorological variables and 46 static ("ancillary") attributes is insufficiently justified. Although the authors state that the variable selection is based on prior studies, the reasoning remains unclear from a reader's perspective. For a data paper, where the choice of variables should be supported by scientific or functional justification. Providing the variable lists used in previous datasets and clarifying whether the present study includes all commonly used variables or only a subset would help readers better understand the design and scope of the dataset.*

  A2. Thank you for pointing out that the selection was not sufficiently justified. We added the following table to the appendix that provides a summary of the variables used in out dataset as compared to the four most relevant and widely-used related datasets. It shows that apart from minor details, our dataset includes all commonly used dynamic and ancillary variables and thus allows for good comparability with the other datasets.

**Appendix B: Comparison with Related Datasets**

**Table B1.** This table compares the variables contained in our dataset to those contained in similar datasets. We matched the raw variables contained in the CAMELS dataset (Addor et al. (2017) and Newman et al. (2015)) as precisely as data availability in our study area permits. This means that e.g. the classes for describing land cover type may vary, although the kind of information is the same. Our dataset also matches other established datasets in this domain rather closely in terms of selection of variables, namely CARAVAN (Kratzert et al., 2023), CAMELS-DE (Loritz et al., 2024) and CAMELS-GB (Coxon et al., 2020). The only substantial difference to all these datasets is that we opted not to include derived climate and hydrological signatures of basins, since any aggregation of our data is optional and would depend on the application. Arguably, since spatially resolved data is more abundant and detailed, it might render "summary statistics" of entire basins less relevant to begin with. In any case, signatures can be readily calculated from the raw data contained in our dataset according to the user's preference. Throughout the table, all listed temporal variables across all datasets are aggregated daily, so "mean" temperature refers to daily mean etc.

| | | Our Dataset | CAMELS | CARAVAN | CAMELS-DE | CAMELS-GB |
|---|---|:---:|:---:|:---:|:---:|:---:|
| Dynamic / meteo. variables | Temperature minimum | ✓ | ✓ | ✓ | ✓ | |
| | Temperature mean | ✓ | | ✓ | ✓ | ✓ |
| | Temperature maximum | ✓ | ✓ | ✓ | ✓ | |
| | Dew point temperature min, mean, max | | | ✓ | | |
| | Potential evapotranspiration | ✓ | | ✓ | ✓ | ✓ |
| | Pot. evapot. (rain corrected) | | | | | ✓ |
| | Daily precipitation sum | ✓ | ✓ | ✓ | | ✓ |
| | Daily precipitation variance | ✓ | | | | |
| | Daily precip. min, mean, median, max, std | | | | ✓ | |
| | Wind components N- and E-ward, min, mean, max | | | ✓ | | |
| | Windspeed | | | | | ✓ |
| | Humidity mean | | ✓ | | ✓ | ✓ |
| | Humidity minimum | | | | ✓ | |
| | Humidity maximum | | | | ✓ | |
| | Shortwave radiation | | ✓ | ✓ | | ✓ |
| | Shortwave radiation min, max | | | ✓ | | |
| | Longwave radiation | | | ✓ | | ✓ |
| | Longwave radiation min, max | | | ✓ | | |
| | Net surface radiation min, mean, max | | | ✓ | ✓ | |
| | Net surface radiation median, std | | | | ✓ | |
| | Surface pressure min, mean, max | | | ✓ | | |
| | Day length | | ✓ | | | |
| | Snow water equivalent | | ✓ | | | |
| Ancillary / static variables | Hydrogeology | ✓ | ✓ | ✓ | ✓ | ✓ |
| | Land Cover | ✓ | ✓ | ✓ | ✓ | ✓ |
| | Soil Type | ✓ | ✓ | ✓ | ✓ | ✓ |
| | Orography | ✓ | ✓ | ✓ | ✓ | ✓ |
| Basin signatures | Hydrological signatures | | ✓ | ✓ | ✓ | ✓ |
| | Climate signatures | | ✓ | ✓ | ✓ | |
| | Gauging station properties | | ✓ | ✓ | ✓ | ✓ |
| | Human influence attributes | | | ✓ | ✓ | ✓ |
| | Simulated hydrologic time series (model output) | | | | ✓ | |
| River gauge data | Catchment discharge | | | | ✓ | ✓ |
| | Catchment-specific discharge | | | | ✓ | ✓ |
| | Water level | | | | ✓ | |

Figure 1: Comparison of variables in our dataset with related datasets.

- *R3. The manuscript mentions "reprojecting and subsampling at the locations of the nodes in this grid," but does not provide methodological details. For a dataset paper, the resampling and reprojection procedures should be described, including the specific interpolation or sampling methods used. Without this information, the processing steps are not sufficiently transparent or*

*reproducible.* A3. We agree that this information is crucial and should be contained directly in the manuscript. We substantially extended section 2.3 (previously 2.1) "Grid Harmonization" to provide more specific detail on all resampling methods. Please also note that we provide the full preprocessing code for transparency and reproducibility. Below we provide the new section:

**Grid Harmonization:**

As a common grid format, we decided to use the grid of the ERA5-Land reanalysis dataset (Muñoz Sabater, 2019; CopernicusClimateChangeService, 2022), which covers the earth's surface with a $0.1° \times 0.1°$ resolution, corresponding to roughly $9km \times 9km$ for the case of our study area. ERA5-Land contains a vast number of meteorological variables and has an hourly resolution, spanning from 1950 onward. It has been widely used and is actively maintained and updated. This means the dataset we provide with this paper remains easily extendable, should a user like to e.g. include an additional meteorological variable in their experiments, extend the study area or increase the temporal resolution. Our study area comprises a total of 7169 grid cells on this grid.

The spatial data originally comes in different formats (vector or grid), projections and resolutions. All data sources were thus harmonized to the grid of ERA5-Land. The first step consisted in converting the maps to a common coordinate system. For the sake of compatibility with ERA5, we decided to use the geographic coordinate system WGS 84. Then for each map separately, polygons covering 0.1▪ in latitude and longitude with the grid cell at the center were extracted. For categorical maps like hydrogeology and land cover, consisting of classes such as "Artificial surfaces: Urban fabric", the fractions covered by each class within each polygon were calculated. This results in a distributional description of class occurrence maximally conserves the original information, as no averaging or other kind of aggregation is necessary. Quantitative information, like e.g. clay content in topsoil, was however aggregated in a final step within the polygon. As a downside to this approach, note that both cases require calculating coverage areas in a geographic coordinate system. This treats the surfaces as flat instead of accounting for the earth's curvature, making the calculations imprecise. We consider this effect negligible here since the surfaces contained in the grid cells are so small that they can be safely considered approximately flat. A more severe limitation is the fact that at high latitudes, polygons defined by a given latitudinal and longitudinal extent become substantially more narrow on the side facing the pole, which also impairs the calculation of area. We could not change the polygons to counteract this effect since they are implicitly dictated by the ERA5 grid. At the moderate latitudes of our study area and especially the small polygons used in the grid, this distortion can still be considered acceptable for the sake of harmonization with ERA5, depending on the application. However, applying this approach in polar regions for example would necessitate an intermediate step of choosing a suitable projected coordinate system for calculating the areas in order to make the distortion explicit and thus better understand its effects. Boundary effects are not an issue with this vector approach as no interpolation is required, however. All maps covered regions well exceeding the study area.

Specifically for the four map types, the hydrogeological map already came in the WGS 84 reference system. The land cover map came in vector format and LAEA coordinate system, so the polygon's coordinates could simply be calculated point-wise using the geopandas Python package (den Bossche et al., 2025) with pyproj (Snow et al., 2025) as a backend. This was followed by calculating the fraction of each class within the polygon described above. The soil map came in LAEA reference system but as a raster. We decided against interpolating as it would not contribute any additional information. Instead, we simply considered each cell in the raster as a separate polygon and calculated the fractions using the vector method described above. The situation was the same for the elevation map, with the added initial step of down-sampling the map from 25 m resolution to 500 m resolution using weighted average resampling implemented in rasterio (Mapbox, 2024).

- *R4. The manuscript does not address several essential aspects related to data quality and reliability. It does not discuss whether resampling introduces information loss, whether the gridding process may generate boundary effects, or whether any variables contain missing values and how such cases are handled. These issues are fundamental for a data paper, as they allow users to assess the reliability of the dataset for their applications.*

  A4. Similar to the comment before, we hope that our modified subsection on grid harmonization now addresses all these valid concerns in a satisfactory manner.

- *R5. The conclusion section is brief and lacks a comprehensive synthesis expected of a dataset paper. A proper conclusion should summarize the dataset's scientific contributions, outline the types of research it can support, and explicitly discuss its limitations.*

  A5. We updated the conclusion section to elaborate more on contribution, supported types of research and are more explicit about the limitations. This is the new conclusions section:

Combining a variety of data sources, we provide the first spatially resolved dataset for multivariate rainfall-streamflow modeling. It covers five entire river basins and it thus particularly suited for large-scale modeling of hydrological processes. With the publication of this dataset, we hope to stimulate further development of spatially resolved, high-resolution hydrological modeling beyond the scope of lumped catchments. Suitable for neural network models as well as conceptual and physical modeling approaches, we hope this dataset will facilitate model comparison and stimulate future development in the spatially resolved domain. Using the same spatial grid as ERA5 as well as daily resolution limits its expressiveness of small scale, e.g. convective events, where higher spatial and temporal resolution would be preferable. If users decide to spatially aggregate our data and want to use derived variables like hydrological or climate signatures as input for their models, they would have to manually compute them from the raw data contained in our dataset. This is especially the case for snow-related variables, as particularly the Southeast of our study area is dominated by snowmelt dynamics. Lastly, the dataset is of only limited use for training training models that are to be deployed world-wide. We focus on a contiguous area in central Europe, which means in turn that the dataset contains only a particular subset of all hydrological dynamics that can be observed.

- *R6. The manuscript frequently emphasizes the dataset's applicability for neural network–based hydrological modeling, yet the Introduction does not sufficiently cite relevant literature or explain the research gap. the authors should include supporting references and more clearly articulate how this dataset connects to and advances existing machine learning hydrology research.*

  A6. We appreciate your feedback on this point. Since we used neural network models in our own research to show that such large amounts of data can be numerically handled, we were biased to bring them up too frequently. We changed the manuscript to make it more clear that any spatially distributed modeling approach is suited for this dataset. We also added recent supporting references regarding both neural-network and physical modeling to substantiate our motivation to publicly release this dataset. Please refer to the difference file for a convenient but concise overview of all the changes in this regard.

**References**

CopernicusClimateChangeService: ERA5-Land Hourly Data from 1950 to Present. Copernicus Climate Change Service (C3S) Climate Data Store (CDS), DOI: 10.24381/Cds.E2161bac (Accessed on 23-Oct-2021), 2022.

den Bossche, J. V., Jordahl, K., Fleischmann, M., Richards, M., McBride, J., Wasserman, J., Badaracco, A. G., Snow, A. D., Roggemans, P., Ward, B., Tratner, J., Gerard, J., Perry, M., Taves, M., Hjelle, G. A., carsonfarmer, Tan, N. Y., Bell, R., ter Hoeven, E., Caria, G., Cochran, M. D., rraymondgh, Culbertson, L., Bartos, M., Chai, C. P., Eubank, N., sangarshanan, Flavin, J., and Rey, S.: Geopandas/Geopandas: V1.1.2, Zenodo, https://doi.org/10.5281/zenodo.18024156, 2025.

Mapbox: Rasterio v1.4.3, Mapbox, 2024.

Muñoz Sabater, J.: ERA5-Land Hourly Data from 1950 to Present. Copernicus Climate Change Service (C3S) Climate Data Store (CDS). DOI: 10.24381/Cds.E2161bac (Accessed on 17-Sep-2024), 2019.

Snow, A. D., Cochran, M., Miara, I., Hoese, D., den Bossche, J. V., Mayo, C., Lucas, G., Cochrane, P., de Kloe, J., Karney, C., Shaw, J. J., Anh, T. Q., Filipe, Ouzounoudis, G., Couwenberg, B., Lostis, G., Dearing, J., Jurd, B., Gohlke, C., Schneck, C., McDonald, D., Taves, M., Itkin, M., May, R., Stewart, A. J., de Bittencourt, H., Little, B., Hugonnet, R., and Rahul, P. S.: Pyproj4/Pyproj: 3.7.2rc1, Zenodo, https://doi.org/10.5281/zenodo.16817340, 2025.

---

## Author Comment (AC2)

Dear Reviewer, we thank you very kindly for the constructive comments and time spent in reviewing the manuscript. We have carefully revised the manuscript according to the comments and suggestions. Below we provide a point-by-point response.

- *R1. The dataset is a compilation of several data sources that are quite well-known and acclaimed in the international hydrological community, however bringing this data together undoubtedly was an effort. At the same time, these sources of data ensure consistency and universality for the resulting dataset under review a priori. It was already mentioned in the above comment, that the regridding procedures are not documented in the manuscript, hence its' consistency still needs validation.*

  A1. We agree that this information should be contained in the manuscript itself rather than just the preprocessing code that we provide. We substantially extended the section on grid harmonization to address these concerns while also discussing limitations more explicitly. Please find below the new content section

  **Grid Harmonization:**

  As a common grid format, we decided to use the grid of the ERA5-Land reanalysis dataset (Muñoz Sabater, 2019; CopernicusClimateChangeService, 2022), which covers the earth's surface with a $0.1° \times 0.1°$ resolution, corresponding to roughly $9km \times 9km$ for the case of our study area. ERA5-Land contains a vast number of meteorological variables and has an hourly resolution, spanning from 1950 onward. It has been widely used and is actively maintained and updated. This means the dataset we provide with this paper remains easily extendable, should a user like to e.g. include an additional meteorological variable in their experiments, extend the study area or increase the temporal resolution. Our study area comprises a total of 7169 grid cells on this grid.

  The spatial data originally comes in different formats (vector or grid), projections and resolutions. All data sources were thus harmonized to the grid of ERA5-Land. The first step consisted in converting the maps to a common coordinate system. For the sake of compatibility with ERA5, we decided to use the geographic coordinate system WGS 84. Then for each map separately, polygons covering 0.1‐ in latitude and longitude with the grid cell at the center were extracted. For categorical maps like hydrogeology and land cover, consisting of classes such as "Artificial surfaces: Urban fabric", the fractions covered by each class within each polygon were calculated. This results in a distributional description of class occurrence maximally conserves the original information, as no averaging or other kind of aggregation is necessary. Quantitative information, like e.g. clay content in topsoil, was however aggregated in a final step within the polygon. As a downside to this approach, note that both cases require calculating coverage areas in a geographic coordinate system. This treats the surfaces as flat instead of accounting for the earth's curvature, making the calculations imprecise. We consider this effect negligible here since the surfaces contained in the grid cells are so small that they can be safely considered approximately flat. A more severe limitation is the fact that at high latitudes, polygons defined by a given latitudinal and longitudinal extent become substantially more narrow on the side facing the pole, which also impairs the calculation of area. We could not change the polygons to counteract this effect since they are implicitly dictated by the ERA5 grid. At the moderate latitudes of our study area and especially the small polygons used in the grid, this distortion can still be considered acceptable for the sake of harmonization with ERA5, depending on the application. However, applying this approach in polar regions for example would necessitate an intermediate step of choosing a suitable projected coordinate system for calculating the areas in order to make the distortion explicit and thus better understand its effects. Boundary effects are not an issue with this vector approach as no interpolation is required, however. All maps covered regions well exceeding the study area.

  Specifically for the four map types, the hydrogeological map already came in the WGS 84 reference system. The land cover map came in vector format and LAEA coordinate system, so the polygon's coordinates could simply be calculated point-wise using the geopandas Python package (den Bossche et al., 2025) with pyproj (Snow et al., 2025) as a backend. This was followed by calculating the fraction of each class within the polygon described above. The soil map came in LAEA reference system but as a raster. We decided against interpolating as it would not contribute any additional information. Instead, we simply considered each cell in the raster as a separate polygon and calculated the fractions using the vector method described above. The situation was the same for the elevation map, with the added initial step of down-sampling the map from 25 m resolution to 500 m resolution using weighted average resampling implemented in rasterio (Mapbox, 2024).

- *R2. The authors state that the dataset is more suitable for distributed hydrological models' testing rather than CARAVAN dataset since it's not lumped. While this is obviously true, some*

*uncertainty is still possible originating in interpolation from the variables' sources and eventually in interpolation on the particular model's grid.*

A2. We included a discussion of potential shortcomings of the employed algorithms in the aforementioned new subsection, see above for details. We chose the ERA5 dataset's grid as a target to unify onto since it is very widely used, continuously updated and offers the possibility to add further variables for other research questions.

- *R3. The name of the manuscript refers to rainfall streamflow modelling, however the domain and especially its' southernmost part of is located in snowmelt runoff area. The authors are advised to address this issue, since no snow-related data is given in the dataset.*

A3. Thank you for pointing this out, we made according changes to the manuscript that are listed in the difference file. When compiling the dataset, we tried to ensure maximum comparability to related datasets (please also refer to our answer A2 to the other reviewer. Hence, we did not include snow specifically, although ERA5 offers several snow-related products which could be integrated into our dataset with the code with provide. We also consider snow-related variables to be somewhat derived rather than raw input variables. In "Hydrological concept formation inside long short-term memory (LSTM) networks" (Lees et al., 2022)for example, the authors use snow depth as a "diagnostic probe" to show that neural network models create physically plausible representations of the world based on "raw" input variables.

**References**

CopernicusClimateChangeService: ERA5-Land Hourly Data from 1950 to Present. Copernicus Climate Change Service (C3S) Climate Data Store (CDS), DOI: 10.24381/Cds.E2161bac (Accessed on 23-Oct-2021), 2022.

den Bossche, J. V., Jordahl, K., Fleischmann, M., Richards, M., McBride, J., Wasserman, J., Badaracco, A. G., Snow, A. D., Roggemans, P., Ward, B., Tratner, J., Gerard, J., Perry, M., Taves, M., Hjelle, G. A., carsonfarmer, Tan, N. Y., Bell, R., ter Hoeven, E., Caria, G., Cochran, M. D., rraymondgh, Culbertson, L., Bartos, M., Chai, C. P., Eubank, N., sangarshanan, Flavin, J., and Rey, S.: Geopandas/Geopandas: V1.1.2, Zenodo, https://doi.org/10.5281/zenodo.18024156, 2025.

Lees, T., Reece, S., Kratzert, F., Klotz, D., Gauch, M., De Bruijn, J., Kumar Sahu, R., Greve, P., Slater, L., and Dadson, S. J.: Hydrological Concept Formation inside Long Short-Term Memory (LSTM) Networks, Hydrology and Earth System Sciences, 26, 3079–3101, https://doi.org/10.5194/hess-26-3079-2022, 2022.

Mapbox: Rasterio v1.4.3, Mapbox, 2024.

Muñoz Sabater, J.: ERA5-Land Hourly Data from 1950 to Present. Copernicus Climate Change Service (C3S) Climate Data Store (CDS). DOI: 10.24381/Cds.E2161bac (Accessed on 17-Sep-2024), 2019.

Snow, A. D., Cochran, M., Miara, I., Hoese, D., den Bossche, J. V., Mayo, C., Lucas, G., Cochrane, P., de Kloe, J., Karney, C., Shaw, J. J., Anh, T. Q., Filipe, Ouzounoudis, G., Couwenberg, B., Lostis, G., Dearing, J., Jurd, B., Gohlke, C., Schneck, C., McDonald, D., Taves, M., Itkin, M., May, R., Stewart, A. J., de Bittencourt, H., Little, B., Hugonnet, R., and Rahul, P. S.: Pyproj4/Pyproj: 3.7.2rc1, Zenodo, https://doi.org/10.5281/zenodo.16817340, 2025.